# Evaluation of DNA Extraction Methods Developed for Forensic and Ancient DNA Applications Using Bone Samples of Different Age

**DOI:** 10.3390/genes12020146

**Published:** 2021-01-22

**Authors:** Catarina Xavier, Mayra Eduardoff, Barbara Bertoglio, Christina Amory, Cordula Berger, Andrea Casas-Vargas, Johannes Pallua, Walther Parson

**Affiliations:** 1Institute of Legal Medicine, Medical University of Innsbruck, 6020 Innsbruck, Austria; mayra.eduardoff@gmail.com (M.E.); christina.amory@i-med.ac.at (C.A.); cordula.berger@i-med.ac.at (C.B.); johannes.pallua@i-med.ac.at (J.P.); 2Department of Public Health, Experimental and Forensic Medicine, Section of Legal Medicine and Forensic Sciences, University of Pavia, 27100 Pavia, Italy; barbara.bertoglio01@universitadipavia.it; 3LABANOF, Laboratory of Forensic Anthropology and Odontology, Department of Biomedical Sciences for Health, Section of Legal Medicine, University of Milan, 20133 Milan, Italy; 4Grupo de Genética de Poblaciones e Identificación, Instituto de Genética, Universidad Nacional de Colombia, 11001 Bogotá, Colombia; lacasasv@unal.edu.co; 5University Hospital for Orthopedics and Traumatology, Medical University of Innsbruck, 6020 Innsbruck, Austria; 6Forensic Science Program, The Pennsylvania State University, State College, PA 16802, USA

**Keywords:** bone DNA extraction, DNA quantification, Ion S5 sequencing, mtDNA sequencing, VISAGE Basic Tool, MiSeq FGx, ForenSeq, SNP sequencing, STR typing and sequencing

## Abstract

The efficient extraction of DNA from challenging samples, such as bones, is critical for the success of downstream genotyping analysis in molecular genetic disciplines. Even though the ancient DNA community has developed several protocols targeting small DNA fragments that are typically present in decomposed or old specimens, only recently forensic geneticists have started to adopt those protocols. Here, we compare an ancient DNA extraction protocol (Dabney) with a bone extraction method (Loreille) typically used in forensics. Real-time quantitative PCR and forensically representative typing methods including fragment size analysis and sequencing were used to assess protocol performance. We used four bone samples of different age in replicates to study the effects of both extraction methods. Our results confirm Loreille’s overall increased gain of DNA when enough tissue is available and Dabney’s improved efficiency for retrieving shorter DNA fragments that is beneficial when highly degraded DNA is present. The results suggest that the choice of extraction method needs to be based on available sample, degradation state, and targeted genotyping method. We modified the Dabney protocol by pooling parallel lysates prior to purification to study gain and performance in single tube typing assays and found that up to six parallel lysates lead to an almost linear gain of extracted DNA. These data are promising for further forensic investigations as the adapted Dabney protocol combines increased sensitivity for degraded DNA with necessary total DNA amount for forensic applications.

## 1. Introduction

DNA extraction is the first and perhaps most important step in any DNA analysis workflow. Particularly in the fields of ancient DNA and forensic genetics, where the samples under investigation are often highly degraded and DNA is present in low copy number, highly efficient DNA extraction methods for skeletal remains are elemental. Although recent ancient DNA studies have proven the possibility of extracting and analyzing DNA preserved in skeletal remains and sediments from up to hundreds of thousands of years [1,2,3,4], the continuous research on new and improved extraction methods indicates the boundaries for efficient DNA isolation and recovery are still to be defined [5,6,7,8,9,10,11,12,13,14,15]. The general principle of DNA extraction from bone powder typically consists of the following steps: The incubation in a lysis buffer in order to chemically break tissue and cell structures is followed by an incubation step in a highly concentrated salt binding buffer that supports the binding of the DNA to silica (e.g., prepared in columns). The DNA is then washed with an ethanol-based solution to minimize inhibitor carry-over and eluted in a low concentrated salt buffer.

The development of new sequencing technologies and library preparation methods allowed moving from PCR-based assays to hybridization protocols and shotgun sequencing. This enabled the capturing of shorter DNA fragments than PCR could amplify. In response, ancient DNA research has focused on developing extraction methods that allowed the isolation and retention of shorter DNA fragments, which are known to be much more abundant than larger fragments in ancient samples [16,17,18]. In 2013, Dabney et al. published a silica-based extraction method that successfully recovered DNA fragments down to 35 bp [11]. More recent publications describe new methods that allow for the recovery of even shorter DNA fragments (≥25 bp) by using a modified binding buffer [13]. Another paper introduced adjustments to the Dabney protocol to optimize recovery (incubation times and temperatures) and introduce a higher degree of automation (for example using magnetic beads instead of silica columns; [14]).

The majority of forensic genetic workflows are still PCR-based and use electrophoretic (CE) fragment sizing, such as the widely applied short tandem repeat (STR) analysis [19] or amplicon-based massively parallel sequencing (MPS), e.g., multiplex-PCR primer tiling assays using smaller amplicons [20,21,22,23,24]. Traditionally, forensic genetics approached to analyze skeletal remains involve lysis protocols including a total demineralization protocol published by Loreille et al. (2007, [25]), which was later optimized and automated, e.g., by Amory et al. (2008, [26]). In the last couple of years however, some forensic laboratories have started to apply alternative extraction and typing methods specifically in highly damaged samples, where PCR-based amplification strategies failed to provide results [27,28,29]. In our hands, the Dabney protocol, originally devised for ancient DNA, provided promising results for the extraction of DNA from skeletal remains in a forensic setting (unpulished data). We adapted this protocol and directly compared its results with the Loreille protocol. In the course of our study, a paper appeared that had a similar research strategy in mind. The authors described the first comparison between Dabney and Loreille extraction protocols in skeletal remains that were burnt to varying degrees [30]. The authors demonstrated the Dabney protocol to be a viable alternative to retrieve full or partial STR profiles from samples burnt at and above 550 °C, while they reported the Loreille protocol to be more effective in better preserved samples. The fact that the Loreille protocol can use higher amounts of bone powder (500 mg to several g) enables higher total DNA yield, while the Dabney was optimized for lower tissue amounts (up to 50 mg). Our study compares both extraction protocols with respect to DNA yield from bone samples of different age (30–2000 years) and describes the performance of various downstream genotyping methods, involving CE-based fragment size analysis (STRs) as well as MPS-based applications (SNPs, STRs and mitochondrial (mt)DNA). In addition, higher DNA bone powder amounts were also tested for the Dabney method. For this purpose, this study compared two different approaches: (i) increasing the bone powder amount (from 50 to 100 mg) and (ii) combining several lysates deriving from 50 mg starting amounts in one or two spin columns.

## 2. Materials and Methods

### 2.1. Samples and Extraction Methods

Four different bone samples offering enough tissue for multiple parallel and consecutive DNA extractions were chosen for the experiments, i.e., a recent, circa 30-year-old bone sample from Austria used for teaching purposes (sample A), two medieval bone samples (between 800 and 1500 years old) excavated from an archaeological site in Volders, Austria (samples B and C; [31]), and a 2000-year-old bone sample from an archaeological site in Sogamoso, Colombia (sample D, [32]). All bones were mechanically cleaned with a scalpel to remove debris from the surface and cut into small pieces for milling. Then, they were chemically cleaned by a series of sequential washes with sodium hypochlorite, water, and ethanol shaken at 300 rpm for 15–20 min. The sampled bone pieces were dried overnight under a laminar flow and milled into bone powder using a MM2 ball mill (Retsch, Haan, Germany). The resulting bone powder was separated into 500, 100 and 50 mg aliquots and subjected to two different extraction methods, from Loreille (Lor) et al. [25,26] and Dabney (Dab) [11]. All bone samples were tested in triplicates for each extraction method and configuration.

#### 2.1.1. Adapted Loreille Method

Total amounts of 500 and 50 mg of bone powder were used for extraction, combined with 6.5 mL of lysis buffer (500 mM of EDTA pH 8.00, 1% N-Laurylsarcosin) and 130 µL of proteinase K (20 mg/mL). In order to directly compare the methods, we decided not to alter the protocol and adapt volumes to 50 mg of bone input. All samples were incubated overnight at 56 °C in a rotary oven with an extra addition of 100 µL proteinase K after 24 h. The lysates were then filtered and concentrated using 30 kDA Millipore Sigma Amicon Ultra Centrifugal Filter Units (Thermo Fisher Scientific, Walthan, MA, USA, herein TFS) for 40 min (centrifugation step at 4500× *g*). Approximately 300 µL of the concentrated lysate were then purified twice using the MinElute (Qiagen, Hilden, Germany) protocol and eluted into 50 µL final volume (two elution steps of 25 µL).

#### 2.1.2. Adapted Dabney Method

Total amounts of 50 and 100 mg of bone powder were tested using this method, joined with 1 mL extraction buffer (450 mM EDTA pH 8.00 and 0.05% Tween 20) and 25 µL of proteinase K (at 10 mg/mL) and incubated for one and/or for two days (depending on the optimization test, see Appendix A) at 37 °C, 56 °C , and a combination of one day at 37 °C and one hour at 56 °C. After incubation, the lysate was combined with 10 mL of the in-house prepared binding buffer (5 M guanidine hydrochloride, 40% isopropanol, 0.05% Tween 20, as in [11]) and 400 µL of 3 M sodium acetate. The solution was transferred into an apparatus composed of a purification column (MinElute) attached to a reservoir (Zymo Research, Irvine, CA, USA) inside a 50 mL Falcon tube and centrifuged at 1500 rpm for 4 min. All tubes were set at 90° and centrifuged again at 1500 rpm for 2 min. The column was transferred to a Qiagen collection tube and centrifuged for 1 min at 6000 rpm. A washing step was repeated twice by adding 700 µL of PE buffer (Qiagen) to each sample prior to a centrifugation step of 30 s at 6000 rpm. The columns were turned for 180° and centrifuged twice to remove all remaining ethanol for 1 min at maximum speed. Finally, the elution was performed in two steps, by adding 25 µL (final volume of 50 µL) of EB buffer followed by 5 min incubation and a centrifugation step of 30 s at maximum speed.

In order to test the Dabney method with higher quantities of bone powder, an alternative protocol was applied. Several aliquots of 50 mg of two bones (bone B for 3 × 50 mg and bone C for 6 × 50 mg) were lysed separately using the Dabney protocol and buffer, but each three replicates were flown through the same column-reservoir apparatus in a sequential way. DNA was eluted from the spin columns in two steps of 25 μL producing the 3 × 50 mg replicates, or, in case of 6 × 50 mg replicates, two elution steps of 12.5 μL and the eluates of two spin columns were pooled together to a final volume of 50 μL. In total, three replicates of 3 × 50 mg and three replicates of 6 × 50 mg bone sample were produced.

### 2.2. Real-Time Quantitative PCR

All DNA extracts were quantified using real-time quantitative PCR (qPCR) following two different approaches. First, a real-time qPCR multiplex based on TaqMan (TFS) chemistry was performed that allows for the simultaneous quantification of one nuclear DNA target and two differently-sized (69 and 143 bp) mtDNA targets (“SDquants”, [33]). Furthermore, an additional mtDNA quantification assay using SYBR Green (TFS) chemistry was performed that targets a 51 bp fragment. After quantification, the triplicates from the same method and conditions were pooled in order to have sufficient volume for all for downstream DNA analyses.

### 2.3. mtDNA Massively Parallel Sequencing

Sequencing libraries were prepared using the Ion Plus Fragment Library Kit (TFS) with 25 µL of extracted DNA for most samples (triplicate pool), except sample A trials (1 µL), which had shown higher mtDNA copy numbers per µL (Appendix A). All samples were prepared following the 100 ng protocol established by the manufacturer. After ligation and purification, all libraries were amplified for 10 cycles according to the manufacturer’s protocol and purified with AMPure XP (Beckman Coulter, Brea, CA, USA) beads with a final elution in 26 µL. After checking the libraries using the Agilent Bioanalyzer High Sensitivity Kit (Agilent, Santa Clara, CA, USA), all libraries followed a primer extension capture-based protocol for mtDNA enrichment as described in [27]. Template preparation and chip loading (Ion 530 Chip, TFS) were performed automatically using the Ion Chef (TFS) and sequenced using the Ion S5 (TFS) with the Ion S5 Precision ID Chef and Sequencing Kit (TFS). Raw data was aligned to the rCRS [34], using the Torrent Suite Server (v. 5.10) TMAP algorithm. Variant calling was performed manually after inspection of all aligned data using the Integrative Genomic Viewer (IGV) [35,36] by two independent analysts. Haplotypes were called following phylogenetic alignment conventions [37,38] according to the International Society for Forensic Genetics (ISFG) recommendations [39] and queried via EMPOP (https://empop.online, [40]) for frequency calculations and haplogroup assignment [41]. The read depth per base and the range of analysis were determined using an in-house developed python script. All aligned data were checked for damage patterns with mapdamage2 [42], which reports the damaged status of a sample, particularly for deamination of C>T at the 5′ end and G>A at the 3′ end of the fragments.

### 2.4. Nuclear DNA STR and SNP Typing

Following the common application in forensic casework, all samples were typed using a standard CE-based STR system (ESX 17; Promega, Madison, WI, USA). The reaction volume was set to 25 μL including: 5 μL of master mix, 2.5 μL of primer mix, and 7.5 μL of PCR-grade water. Whenever possible, 500 pg of DNA were used for amplification, otherwise the maximum volume of 10 μL was utilized. PCR settings were as follows: initial denaturation at 96 °C for 2 min, 30 cycles of 94 °C for 20 s, 59 °C for 2 min and 72 °C for 30 s, final elongation at 72 °C for 5 min and 60 °C for 10 min. Fragments were separated using a 3500 Genetic Analyzer (TFS). All data was analyzed using GeneMapper v. 1.2 (TFS).

In order to assess the performance of the different extracts with an MPS-based SNP assay, all samples were analyzed with the AmpliSeq VISAGE Basic Tool for Appearance and Ancestry (herein VISAGE BT), according to the protocol described in [43] with small alterations. This panel comprises 153 SNPs of which 41 are used for eye, hair, and skin color prediction [44,45,46] and 115 SNPs for inferring the biogeographical ancestry (3 redundant SNPs between the panels). DNA extracts were subjected to a pre-library AMPure XP bead clean-up and also processed without clean-up, for which 6 µL of the DNA extract were combined with 2X AMPure XP beads and purified following the manufacturer’s protocol. A total of 6 µL were used for library preparation for most of the samples, except for sample A extracts, which presented higher DNA quantification results, therefore only 1 µL was used (Appendix A). All libraries were prepared manually using the Ion AmpliSeq Library Kit 2.0 (TFS) and followed the low DNA input protocol with increased PCR cycles (27 cycles) as suggested in [43] and eluted in 25 μL. Final libraries were quantified with the Ion Library TaqMan Quantitation Kit and pooled equimolarly at 30 pM when possible, or used undiluted. Automated template preparation and Ion 530 chip loading were performed with the Ion Chef and sequencing with the Ion S5 using the Ion S5 Precision ID Chef and Sequencing Kit (all TFS). Raw data were aligned to hg19 human genome assembly by the Torrent Suite Server (v. 5.10) TMAP aligner and the HID_SNP_Genotyper plugin v.5.2.2 (TFS) was used to retrieve genotypes and read depth (among other information) per marker.

Furthermore, 12 of the pooled DNA extracts (triplicates) were also typed with the ForenSeq DNA Signature Prep Kit (herein ForenSeq; Verogen, San Diego, CA, USA) following the manufacturer’s recommendations and analyzed in a MiSeq FGx instrument (Verogen) [23]. ForenSeq allows for both SNP and STR typing including a total of 152 markers (58 STRs and 94 SNPs). A pre-library preparation purification step, as described above, was also performed to remove possible carry-over inhibitors. All data were analyzed using the ForenSeq Universal Analysis Software v. 1.3.6897 (Verogen).

## 3. Results and Discussion

A total of 48 DNA extractions from bone and 76 downstream genotyping reactions were performed to fulfill all tests considered in this study. Extracted DNA from bone samples was quantified by real-time qPCR and analyzed using Primer Extension Capture mtDNA MPS (*n* = 16), nuclear SNP and STR MPS (VISAGE Basic Tool (*n* = 32), and ForenSeq (*n* = 12)) and CE-based STR typing (ESX 17 system, *n* = 16). Real-time qPCR results are presented here as mean values of the duplicate qPCR and the triplicate extraction experiments, whereas for downstream genotyping, the triplicates were pooled before the protocols.

### 3.1. Nuclear and mtDNA Quantification

As a general and expected observation, larger amounts of input material (bone powder) resulted in higher DNA amounts. Therefore, the Loreille protocol using 500 mg (Lor 500) of bone powder yielded higher absolute DNA quantification results compared to most Dabney (Dab) protocols (Appendix A). A few exceptions were observed for the mt51 bp target in DNA extracts from sample B, where Dab-3 × 50 mg showed a higher value as well as sample D, where both Lor-500 and Dab-100 presented similar values. However, the comparison between protocols using such different amounts of bone powder input does not reflect the method’s efficiency. Therefore, both mtDNA and nuclear DNA quantification results were also normalized by the amount of bone powder used (Figure 1). This resulted in similar normalized quantification results for the 143 bp mtDNA target within the same bone sample, whereas the shorter mtDNA targets (69 and 51 bp) as well as the nuclear target (70 bp) showed higher normalized DNA quantification results for the Dab protocol in all bone specimens (Figure 1).

Interestingly, even the minor differences in the short amplicon sizes provided higher normalized quantification results, e.g., mt51 vs. mt69 data (Figure 1). We note, that because the mt51 is a SYBR Green assay, it is more prone to unspecific signal when compared with TaqMan based assays (mt69). Statistic t-tests were performed comparing the triplicate results for sample B different tests of the 2016 extraction trials, Dab-100 mg was significantly different from Dab-50 mg (*p*-value < 0.05), indicating the assay’s higher efficiency with less bone input. Furthermore, while for the tests using 56 °C (Dab-50 mg vs. Dab-50 mg-1d), the differences were statistically significant (*p*-value < 0.05), all the other comparisons with the 37 °C (Dab-50 mg-37 vs. Dab-50 mg-1d-37) and between the 37 °C and 56 °C were mostly non-significant. Two exceptions were seen when comparing Dab-50 mg-37-1d with Dab-50 mg for the nuclear quantification target and the shorter mtDNA target (*p*-values < 0.05); however, when comparing using all the data combined, the difference was also non-significant.

In order to evaluate the increase of DNA yield with the Dab method, we tested alternative protocols by doubling the bone powder amount using the same protocol (Dab-100) and by combining three (Dab-50-3 × 50 mg) or six (Dab-50-6 × 50 mg) 50 mg lysates. The results from Lor-500/Lor-50, Dab-100/Dab-50, Dab-3 × 50/Dab-50, and Dab-6 × 50/Dab-50 were directly compared to understand the gain/loss of the respective method and evaluate recovery efficiencies (Appendix A). Combining 3 and 6 × 50 mg Dab lysates brought quantification values close to the proportional theoretical values (averages of 0.074–10× loss, 2.77–3× gain, and 5.82–6× gain). These results suggest an almost linear gain of DNA with the amount of input bone powder used in parallel assays, which was assessed by performing linear regression analyses for all samples B and C triplicates and compared with the respective theoretical results. Expected lines were extrapolated from the mean values of Lor-500 mg and Dab-50 mg, respectively (Appendix A). Due to the different extraction dates of Sample B, the Dab-50 mg triplicates used to estimate an ideal 3× gain were the ones performed in 2020 as the Dab-3 × 50 mg replicates. Unfortunately, for the Loreille method, different extraction dates for Lor-50 and Lor-500 mg were used. Differences in absolute quantification values between extracts using the same conditions (sample B Dab-50 mg-1d-37 and Dab-50 mg-1d-37 *) but performed in 2016 and in 2020 might have been caused due to storage conditions and poor DNA preservation. Indeed, a test using sample D was performed to assess this hypothesis and the same observation was seen (unpublished data). Although the Lor ratio comparison showed a higher loss than the expected 10 fold, the overlap between observed and expected lines points towards a linear and proportional gain for both Lor 10× loss and Dab 6× gain (average R^2^ of 0.974 and 0.962, respectively). Dab 3× gain showed a slight decrease in the observed line in two targets; however, no significant difference between lines was observed (average R^2^ = 0.883; Appendix A). Doubling the amount of bone powder in a single Dab assay (Dab-100 mg) turned out to be inefficient (Appendix A) as previously demonstrated [47], confirming the efficiency and optimization of the Dab protocol for 50 mg lysates (average R^2^ = 0.092).

The two extraction methods were directly compared using recent DNA extracts of Sample B: the Dab-50–1d-37* and Lor-50 protocols. On average the Dab protocol outperformed the Lor method with 2.2× higher DNA concentration values. We evaluated the practical consideration, how many Dab-50 lysates would need to be combined to compensate for the increased input in the Lor-500 method. As a result, an average of 6 × 50 replicates (300 mg) using the Dab method are sufficient to reach the values obtained for Lor-500 for sample B (Appendix A). In fact, the results for the different quantification targets were ranging from 10× for mt143bp to 2× for mt51bp, which is expected since the Dabney method is optimized for the recovery of shorter DNA fragments. For samples A and D, on average only four Dab-50 lysates (200 mg) would be required to compensate the Lor-500 results.

Aiming to understand the level of degradation, an index was determined for the samples and trials by dividing the quantification results from the short mtDNA TaqMan target (69 bp) by the longer target (143 bp, Appendix A) All replicates from samples B, C, and D, independently of the method, showed a degradation index > 1, indicating degradation as expected for bone remains with 800–1500 and 2000 years of age. Higher values were obtained for the Dab-100 replicates of samples B and D, followed by the Dab-50 mg results. Newly extracted Dab replicates of sample B also showed high degradation values (average index of 8.6), while Lor-500 replicates showed consistent lower degradation values for all samples. Such results corroborate findings of past studies that describe the Dab method to be optimized for shorter DNA fragment recovery.

The observed trend of higher DNA yields per mg tissue, particularly for the shorter quantification targets, agrees with previous Dabney studies [11,14]. Recently, a new extraction method with an altered binding buffer was published, that was aiming to recover even shorter DNA fragments [13]. However, we did not consider this option in our experimental setup, as it was described to be more susceptible to inhibitor carry over, which is a crucial factor in forensic genetic investigations; furthermore, the alignment of shorter sequences is still very complex.

### 3.2. MtDNA Sequencing

Full coverage for the entire control region was obtained for all tested samples using the PEC method. Mean read depth per position for the control region varied from 7740 to 3201 for sample A, from 3091 to 740 for B, 3870 to 1694 for C, and 5367 to 2044 for sample D (Appendix A). Most of the investigated samples presented reads in other regions of the mtGenome, thus increasing the possible range for haplotype distinction (Appendix A,). The entire control region represents only 6.8% of the mtGenome, and all tested samples yielded reads at least covering 15.42% of the mtGenome with 3× coverage (Appendix A). The Lor-500 method reached higher percentage of covered mtGenome only for sample A (99.65% with 3× coverage) when compared to other methods (93.45% and 88.25% for Dab-100 and Dab-50, respectively, Appendix A, Figure 2A). Both samples B and D showed higher percentages of 3× covered mtGenome for the Dab methods (41.6% and 60.93%, respectively).

Interestingly, even though the Dab-3 × 50 mg trial showed similar coverage values as the Dab-50 trial extracted at the same date (Dab-50 mg-1d-37 *), the extraction trial with 6 × 50 mg of bone powder (amounting to 300 mg) showed an increase of 32.8% of covered mtGenome to the other recent trial of Dab-50–1d-37/56 (Appendix A), reaching 90.8% mtGenome coverage. This indicates that this protocol alteration provides an efficient method for primer extension capture MPS. Furthermore, by normalizing the percentage of covered mtGenome per amount of bone powder [mg] used, the efficiency of each method can be visualized in an easier manner (Figure 2A, Appendix A). The Dab-50 extraction showed higher normalized covered mtGenome coverage than the remaining methods for all tested samples. Even though in absolute values most of the samples present higher read depth for the control region obtained with Lor-500 mg (Appendix A), the normalized values of read depth by the total number of reads showed an increased number of reads targeting the mtDNA control region for the Dabney trials. Indeed, the Dab-50 mg-1d-37 reached the highest percentage of mtDNA target reads (22%) when compared with other methods for sample B.

As the mtDNA sequencing method chosen here is a capture-based assay, relevant information can be retrieved from the number of reads per read length, which helps inferring the actual size of the fragments recovered from extraction. Read length density plots were drawn for all extraction trials (Figure 2B and Appendix A) and depict a shift in read length from the Lor-based extractions (mean and median read length of 150 and 142 bp, respectively) to the Dab-based extractions (mean and median read length of 132 and 128 bp, respectively). A gain of recovered fragment sizes ranging from 35–100 bp is visible in the Dab-based extractions (Figure 2B and Appendix A). Considering that the Dabney protocol has been designed and optimized within the ancient DNA community, which prioritizes the recovery of short and highly degraded DNA fragments, a better recovery proportion of fragments around 50–120 bp (±adapters and barcodes) than >120 bp was expected (Figure 2B and Appendix A). Total and informative sequence content for all trials were calculated following [13], but adapted to Ion Torrent sequencing strategy (single end), and normalized to the best method (Appendix A). In addition, trials using more than 50 mg of bone powder were corrected in proportion (e.g., for Lor-500 the values were divided by 10). Both total and informative sequence values were highest for Dab-50 followed by Dab-100 with 0.6 on average and Lor-500 reaching only 0.08 on average (Figure 3, average values for sample B). When plotting all values for the earlier optimization trials performed for sample B (Appendix A), the highest values were obtained for Dab-50 mg-1d-37. These results agree with past publications stating the disadvantages of increasing temperature on DNA preservation [9] and with our observation of a higher percentage of target reads for Dab-50 mg-1d-37 when compared with other trials (average of 15.68% for Dab-50 mg trials and 18.58% for Lor-500 mg). Interestingly, while Lor-500 underperformed with and without correction (0.016 and 0.08, respectively), the Dab trial with 150 mg (Dab-3 × 50 mg-1d-37) still reached 0.097 after the correction. In addition, a direct comparison between the more recent extraction trials with the same bone input showed that Dabney trials reached 39× more informative sequence content than the Lor-based did.

Higher GC content in shorter fragments has been associated with the Dabney extraction method [13] when compared to other ancient DNA extraction methods. Although higher variation in GC content was observed in shorter reads, probably due to adapter dimer formations, no clear tendency towards higher GC content was noted for both methods (Appendix A). Another parameter, that can shed light on the quality of DNA recovered, is the analysis of the frequency of DNA damage found at the extremities of the reads. Previous ancient DNA studies have shown higher prevalence of C>T and G>A substitutions at the 5′ and 3′ end of the reads [48,49,50], but a recent study showed differences in damage frequencies in the interior of the sequences between extraction methods [13]. Indeed, better performing methods lead to higher recovery of DNA fragments, particularly single strand breaks. Although a visible increase in damage can be observed in the extremities (C>T at 5p and G>A at 3p, particularly, Appendix A), the damage variation in inner positions is lower but more variable between methods (Pairwise Wilcoxon test, Bonferroni correction, Appendix A). This observation could be biased by the library preparation method used, since poorer performance for double stranded library preparation methods has been observed with shorter and more damaged DNA fragments [13,51,52,53]. Even though no consistent significant differences were found between methods for both damage patterns (G>A and C>T) at the beginning of 3p and 5p reads, in the center of the reads (>25 bp) Lor-500 showed significantly less damage than most other methods for sample B (Pairwise Wilcoxon test, Bonferroni correction, Appendix A). A reason for this result could be because the Dabney protocols allow for the extraction of shorter and probably more damaged DNA fragments. The Dabney trial with 6 × 50 mg of sample C showed an increased number of damage occurrences both at the beginning as well as in inner positions of the reads, which comes in contrast with Dab-50 mg-1d-37/56 that showed consistently the lower variable number of damage occurrences (Appendix A). The frequency of occurring damages C>T and G>A was stable for both extremities and inner positions for sample A and showed almost no significant differences between methods (*p*-value > 0.05). On the contrary, damage frequencies seem to be consistently higher for the Dabney methods, except for C>T transitions at 5p. As expected, the younger bone sample (A) showed less damage than the other three bones (Appendix A).

Observed haplotypes were found concordant for all replicates of the same bone sample, within the obtained sequencing ranges. Haplogroups were assigned manually using Phylotree v.17 and confirmed using EMPOP’s haplogrouping tool (https://empop.online/). A list of haplotypes, frequencies, and haplogroups is provided in Appendix A.

### 3.3. Nuclear DNA–STR Typing and SNP Sequencing

Three different kits were used to assess the performance of the different extraction methods with nuclear DNA targets and PCR based techniques: a CE STR typing kit commonly used in forensic routine workflow-ESX 17 (*n* = 16), a SNP panel using Ion MPS technology–VISAGE BT (*n* = 32, 16 with and 16 without pre-library purification)–and finally, a panel combining SNP and STR sequencing on the MiSeq–ForenSeq (*n* = 12). Here, all three approaches were tested for the majority of the extraction trials (with the exception of the ForenSeq DNA signature prep kit that was not tested for the 2020 extraction trials). The percentage of called loci was divided by milligram of bone powder used to normalize the results. Overall, Dabney extracted methods showed a better normalized performance for all three approaches reaching even full profiles for the recently extracted trials of sample C (Figure 4). Furthermore, when calculating the percentage of target reads, we obtained higher values for the Dabney methods in all bone samples. In particular, sample B showed the highest percentage (15.7%) in trial Dab-50 mg-1d-37 (2016 extractions), corroborating previous observations. Even though when analyzing the earlier extracted samples, better absolute results are obtained for Lor-500, improved results for sample B can be obtained with the altered Dabney-3 × 50 mg protocol. Indeed, Dab-3 × 50 showed an increase of 2.3× more loci called than Dab-50 mg-1d-37 * for the VISAGE BT panel. According to recently published studies [43,54] targeted MPS methods were proven less tolerant to inhibitors as standard CE-based STR typing panels. Considering the VISAGE AmpliSeq panel, six extraction trials (pooled triplicate extracts) for samples B, C, and D showed performance increases with the extra purification step (B-Dab-100 mg, B-Dab-50 mg-1d, B-Dab-3 × 50 mg-1d-37, C- Dab-50 mg-1d-37/56, C- Dab-6 × 50 mg-1d-37, and D- Dab-50 mg, Appendix A), with gains of 4, 25, 29, 28, 63, and 19 loci called, respectively. However, the clean-up step showed no visible increase in performance for the remaining extracted samples or even a slight decrease in performance, which can be explained by DNA loss during purification (Appendix A). Nine samples (56%) showed an increase in mean read depth with a pre-PCR purification step. Samples A-Lor-500, B-Lor-50, B-Dab-100, and B-Dab-50-37 showed a moderate increase of 3.3–10 × in mean read depth, while samples D-Dab-50, B-Dab-50-1d, B-Dab-3 × 50-1d-37, C-Dab-50-1d-37/56, and C-Dab-6 × 50-1d-37 presented an increase of 13.8–163.6× in mean read depth with the extra purification (Appendix A).

Previous in-house work on bone samples has shown ForenSeq to be sensitive to inhibition carry over from bone extraction methods (unpublished); however, further optimization of the product has taken place that was not yet tested in-house. In addition, a recent publication has described ForenSeq as very sensitive to inhibitor presence [54]. Even though a pre-library purification was performed for all ForenSeq libraries, when compared with the remaining panels, the ForenSeq still underperformed. Such results clearly pinpoint the importance for further optimization of buffers and other components of both AmpliSeq and ForenSeq library preparation kits, particularly for forensic use. Interestingly, the results from the ESX 17 STR-CE based kit were all achieved without a pre-purification step, which clearly highlights the advantages of using a technology that has undergone several years of optimization. CE-STR results followed a similar trendline to the VISAGE BT AmpliSeq panel with the Dabney extracted samples overperforming the Loreille prepared samples in the vast majority (except the Dab-100 mg for sample B). Furthermore, our results agree with a previous study that pinpoints Dabney as a good alternative extraction method for very degraded samples and CE-STR typing [30].

## 4. Conclusions

Although the establishment of the Dabney-based extraction methods has been accomplished in several ancient DNA research laboratories since 2016, the adoption of new protocols in forensic laboratories seems to be a slower and more laborious process. Any new method requires rigorous validation tests to assess the overall performance, advantages, and possible caveats, in order to evaluate its range of applications within forensic workflows. Even though Dabney-based extraction has been proven efficient, particularly with highly degraded/ancient DNA samples [11,14], forensic practitioners may argue that its limited bone powder input restricts the absolute DNA yield, which is ultimately needed for further downstream applications. Furthermore, while the main targets of the ancient DNA community are shorter DNA fragments, the variety of DNA typing methods in forensic genetics still depends heavily on PCR-based methods, both for CE and MPS assays and thus, requires relatively large DNA fragments for successful results. Indeed, while SNP typing strategies imply amplification of short fragments (minimum of 120 bp), STR CE fragment length typing uses amplicons up to 300 bp. Nevertheless, some forensic teams are already applying capture-based techniques for cases where no successful results were obtained using standard PCR-based methods [27,29]. Particularly for cases involving highly degraded DNA, such as burnt samples, the Dabney extraction protocol was proven a valuable tool [30].

This study provides a side-by-side comparison of Dabney- and Loreille-based DNA extraction methods using four different aged bone samples. The results highlight the efficiency of the Dabney-based methods and in particular the Dab-50 mg-1d-37. For most samples the Dabney-based protocols outperformed Loreille-based protocols after correcting for bone powder used in both DNA yield and typing results. In addition, a direct comparison between Dab-50 and Lor-50 indicates that the Dabney method provides higher DNA yields. The overall analysis of the read lengths showed an increase in shorter fragments for the Dabney methods, which is consistent with previous research [11,14]; however, the Dabney methods also showed the best results in terms of informative sequence content (fragments > 35 bp) and percentage of targeted reads, indicating that the shorter fragments were not caused by extraneous DNA or adapter-dimer formations. Furthermore, the Dabney methods showed generally higher frequencies of damage patterns. However, only sample C presented consistently significant differences between protocols. Nevertheless, the Loreille-based method yielded higher overall DNA amounts as this method tolerates higher sample input amounts. Therefore, the Loreille method is preferred in cases where heavy DNA degradation or sample preservation is not an issue.

Finally, we describe an alteration to the Dabney protocol to increase DNA yield by sequentially purifying parallel DNA lysates from the same bone specimen. This strategy compensates for the sample input limitation that Dabney-based protocols showed. This would be particularly important for samples containing highly degraded DNA that allow parallel sampling. Thus, higher DNA yields can be achieved by maintaining the option to recover very short DNA fragments as evident in severely degraded DNA.

## Figures and Tables

**Figure 1 genes-12-00146-f001:**
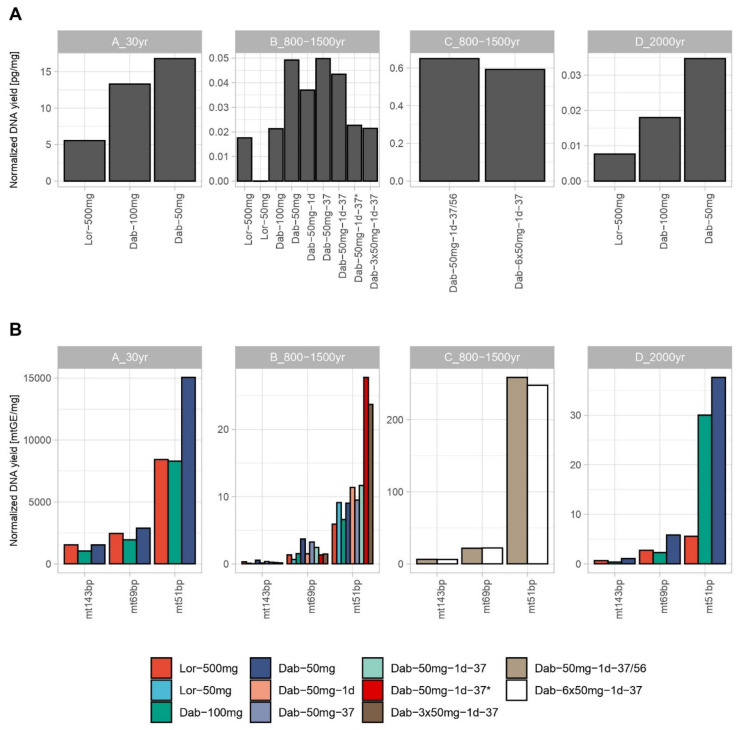
Average DNA yield obtained for each triplicate set of samples per bone normalized to the amount of bone powder (mg) used for extraction for the nuclear quantification target (**A**) and the three differently sized mitochondrial DNA targets (**B**). All extraction trials performed are plotted per sample (A–D). Samples Lor-50 mg, Dab-50 mg-1d-37 *, Dab-3 × 50 mg-1d-37, Dab-50 mg-1d-37/56, and Dab-6 × 50 mg-1d-37 were extracted and quantified in 2020 (while the remaining samples were extracted in 2016 and quantified in 2016 and 2019). As six different extractions were prepared using the same protocol (Dab-50 mg-1d-37), the * symbol represents the average of the replicates extracted in 2020. This designation is adopted all throughout the manuscript.

**Figure 2 genes-12-00146-f002:**
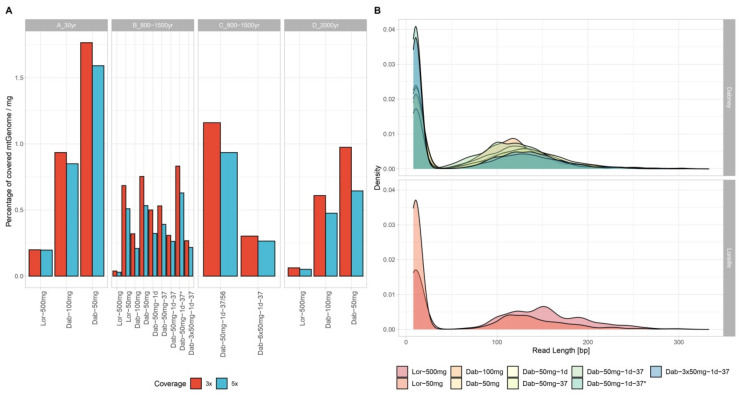
(**A**) Percentage of covered mtGenome per milligram of bone powder used with at least 3 and 5 reads per sample and extraction trial using a primer extension captured based assay. (**B**) Density (number of reads of a certain size in proportion to the total number of reads) plots of read length distribution per sample and extraction trial.

**Figure 3 genes-12-00146-f003:**
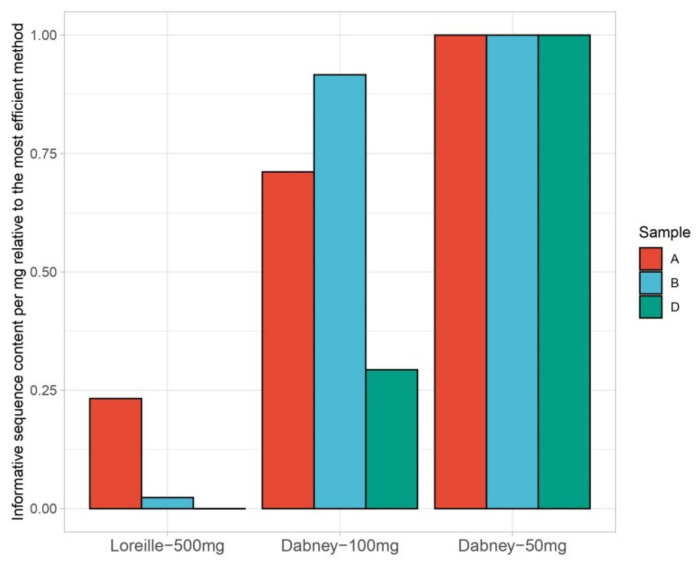
Informative sequence content normalized to the most efficient method (Dabney-50 mg) for samples A, B, and D considering only Lor-500 mg, Dab-100 mg, and Dab-50 mg methods (for sample B, the mean of all earlier extraction trials with 50 mg was calculated). All results were normalized according to the amount of bone powder input (mg).

**Figure 4 genes-12-00146-f004:**
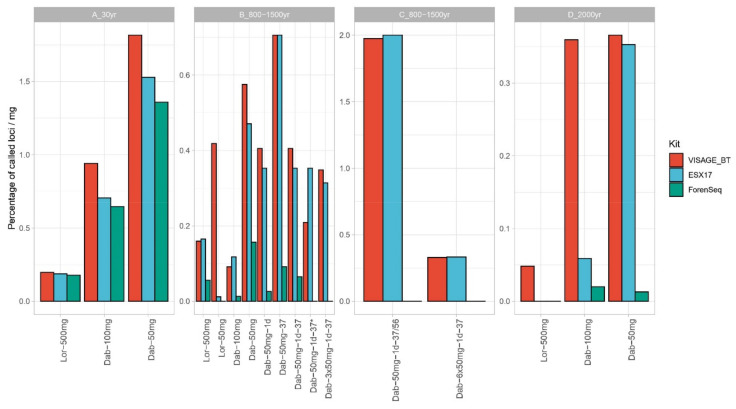
Percentage of called nuclear DNA loci per amount of bone powder input for all samples and extraction trials using VISAGE Basic Tool (153 SNPs in total), ESX 17 (17 STRs in total), and ForenSeq (94 SNPs and 58 STRs). Samples Lor-50 mg, Dab-50 mg-1d-37 *, Dab-3 × 50 mg-1d-37, Dab-50 mg-1d-37/56, and Dab-6 × 50 mg-1d-37 were not typed using the ForenSeq panel because this analysis was performed before these extraction dates.

## Data Availability

The data presented in this study are available in article and Appendix A.

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
