# Peer review of "Evaluation of DNA Extraction Methods Developed for Forensic and Ancient DNA Applications Using Bone Samples of Different Age"

_genes, 2021, doi:10.3390/genes12020146_

Round 1

Reviewer 1 Report

Xavier et al. present a comparison of the bone DNA extraction method most widely used in the ancient DNA community and a widely used method from forensic work. Importantly, they use a statistically meaningful number of comparisons, which is not always the case I such studies. Introduction and M&M part are both clear, the Results & Discussion part is unfortunately sometimes a bit lost in too many details that are described in a rather verbose way. I think that part would benefit from careful editing, just to make it easier for the reader to follow. At the moment the presentation distracts a bit from the quality of the results.

I also have some minor comments, the authors should take into account:

The authors should provide g rather than rpm in the methods section

It is not clear whether they checked DNA quantities before library construction. One can infer from the text that they probably did, but it would be nice if they explicitly mentioned it and provided the data.

When the authors compare performances of different extraction protocols, they need to apply a correction for multiple testing.

The Dabney 100 mg approach not only does not result in DNA gain, but actually results in a DNA loss, well in line with the results by Rohland, Siedel and Hofreiter 2010, Mol Ecol Res.

I suggest that the authors use mapdamage for assessing damage patterns, as the outputs one obtains are much more informative than the current Fig. S6. Generally, some of the supplementary figures are non-intuitive in their presentation of the underlying data and should be revised.

Author Response

We would like to thank both anonymous reviewers for their useful comments! We have addressed all comments and provide responses below.

Reviewer 1

Xavier et al. present a comparison of the bone DNA extraction method most widely used in the ancient DNA community and a widely used method from forensic work. Importantly, they use a statistically meaningful number of comparisons, which is not always the case I such studies. Introduction and M&M part are both clear, the Results & Discussion part is unfortunately sometimes a bit lost in too many details that are described in a rather verbose way. I think that part would benefit from careful editing, just to make it easier for the reader to follow. At the moment the presentation distracts a bit from the quality of the results.

We would like to thank Reviewer 1 for the informative remarks and constructive suggestions. We have streamlined the text in the Results and Discussion section for easier reading. This affected mostly the quantification section (please see alterations highlighted in yellow). We decided to keep the mitochondrial section of that part in its original length (with minor modifications), because the manuscript is intended for the “Mitochondrial DNA” Special Issue of the journal and thus deserves a more elaborate presentation.

I also have some minor comments, the authors should take into account:

The authors should provide g rather than rpm in the methods section

The rpm values stated in the materials and methods section for the Dabney adapted protocol are based on reference 11 and the MinElute protocol (Qiagen).  In order to avoid confusion for a reader when comparing the materials and methods with the reference and protocol, we opted to keep all values in rpm (which are also widely used throughout the scientific community).

It is not clear whether they checked DNA quantities before library construction. One can infer from the text that they probably did, but it would be nice if they explicitly mentioned it and provided the data.

All replicates were quantified before pooling, an average value was considered for downstream library preparation. We have now produced a table with the average DNA quantity for all DNA typing methods (please see new Supplementary Table S2 and lines 150 and 185).

When the authors compare performances of different extraction protocols, they need to apply a correction for multiple testing.

Indeed multiple test comparisons were performed for damage frequency between different extraction tests, for which a Bonferroni correction was applied (see lines 368 and 374).

The Dabney 100 mg approach not only does not result in DNA gain, but actually results in a DNA loss, well in line with the results by Rohland, Siedel and Hofreiter 2010, Mol Ecol Res.

Thank you for this information, we have updated line 257 and added the reference 47.

I suggest that the authors use mapdamage for assessing damage patterns, as the outputs one obtains are much more informative than the current Fig. S6. Generally, some of the supplementary figures are non-intuitive in their presentation of the underlying data and should be revised.

In fact, we’re using mapdamage data to produce figure S6. We decided to present the data in a graphical form because we were interested to visualize the damage frequency not only in the first or last 25 bp, but also within the reads. We think that a graphical representation is more intuitive here as a large table.

Reviewer 2 Report

The paper is not very novel. And more extraction methods could have been explored and compared. But I think that the forensic community would greatly benefit by implementing more and more powerful methods developed in ancient DNA research. So I think that this paper could be of interest for that community, since it properly compares methods used in different fields in a meaningful set of samples.

I would also suggest to add a couple more specimens if possible. While I appreciate that the variability in DNA yield is very high and a proper thorough time stratification is hard to achieve, this would have the additional benefit of providing some more replication. In fact, the authors pooled all replicates, which hides somehow the variability and prevents proper quantitative comparisons. While I understand the need of doing it to save money, increasing even just a bit the number of specimens would give a better idea of the variability. At the moment, the paper has just one 30yr and one 2000yr case. The only condition with two cases is 800-1500yr. To claim that the paper compares specimens from different ages, a bit more should be done I think.

Minor importance:

  • the background section is good and exhaustive. I think that it would have been proper to cite some other "DNA extraction comparison" papers, such as Rohland and Hofreiter 2018, and others.
  • labels and legends in plots are often too small and some (figure 2 and 3) very hard to read. I also think that in some cases tables would have been clearer.
  • Figure 4: plots should be in the same scale.

Author Response

We would like to thank both anonymous reviewers for their useful comments! We have addressed all comments and provide responses below.

Reviewer 2

The paper is not very novel. And more extraction methods could have been explored and compared. But I think that the forensic community would greatly benefit by implementing more and more powerful methods developed in ancient DNA research. So I think that this paper could be of interest for that community, since it properly compares methods used in different fields in a meaningful set of samples.

I would also suggest to add a couple more specimens if possible. While I appreciate that the variability in DNA yield is very high and a proper thorough time stratification is hard to achieve, this would have the additional benefit of providing some more replication. In fact, the authors pooled all replicates, which hides somehow the variability and prevents proper quantitative comparisons. While I understand the need of doing it to save money, increasing even just a bit the number of specimens would give a better idea of the variability. At the moment, the paper has just one 30yr and one 2000yr case. The only condition with two cases is 800-1500yr. To claim that the paper compares specimens from different ages, a bit more should be done I think.

We respectfully disagree with the reviewer’s notion. As Reviewer 1 points out in their first comment, “importantly, they use a statistically meaningful number of comparisons, which is not always the case in such studies.”, we have put a lot of effort in finding enough biological material for parallel testing from extremely diverse age categories. We do not claim that we provide data over a long timeframe, which would then require more sampling.

Our intention is to compare two extraction techniques that are frequently used in the fields of forensic and ancient DNA research and our statistical evaluation demonstrates that this goal was reached with the number of samples used.

This direct comparison is novel for bone specimens; to the best of our knowledge, such direct comparison has not been presented before.

The pooling of samples was not done to save money, although this could be some motivation, the pooling was performed to increase the sheer amount of biological material for downstream genotyping. This is reflecting a real-world routine scenario. Our data show that the performance of the Dabney extraction protocols suffers from an excess of biological material (bone powder), hence, a suitable and successful method to overcome this limitation is to perform parallel extractions that are then pooled for downstream genotyping. This is what we demonstrate with our experiments.

We also show – and that is also novel – that the pooling of up to 6 times of the input material leads to proportional output increase. Hence, pooling of parallel Dabney extracts not only outperforms Loreille protocol but also offers an alternative to enrich absolute DNA amount if needed for the application.

We consider this an extremely important and practical solution for routine forensic casework, which is one of our main tasks and fields of expertise and desired by the forensic comunity.

Minor importance:

  • the background section is good and exhaustive. I think that it would have been proper to cite some other "DNA extraction comparison" papers, such as Rohland and Hofreiter 2018, and others.

We believe the reference in question (Rohland and Hofreiter, Biotechniques) has been recently re published online in future-science.com but refers to an older paper from 2007 (see PDF in the following website https://www.future-science.com/doi/10.2144/000112383 ).This reference has already been included in our manuscript (see reference 9 presenting the same title and doi number). Other DNA extraction comparison papers have been included (please see references: 8, 13 and 14).

  • labels and legends in plots are often too small and some (figure 2 and 3) very hard to read. I also think that in some cases tables would have been clearer.

We have now increased the size of the labels in figures 2 and 3.

  • Figure 4: plots should be in the same scale.

We understand that identical Y-scales make data more comparable, however, this would be at cost for the readability of the figures. Our main goal was the comparison between different extraction tests within the same sample, because this is the relevant part of our study.